# SKILL DECISION TRANSFORMER

## ABSTRACT

Recent work has shown that Large Language Models (LLMs) can be incredibly effective for offline reinforcement learning (RL) by representing the traditional RL problem as a sequence modelling problem (Chen et al., 2021; Janner et al., 2021). However many of these methods only optimize for high returns, and may not extract much information from a diverse dataset of trajectories. Generalized Decision Transformers (GDTs) (Furuta et al., 2021) have shown that by utilizing future trajectory information, in the form of information statistics, can help extract more information from offline trajectory data. Building upon this, we propose Skill Decision Transformer (Skill DT). Skill DT draws inspiration from hindsight relabelling (Andrychowicz et al., 2017) and skill discovery methods to discover a diverse set of *primitive behaviors*, or skills. We show that Skill DT can not only perform offline state-marginal matching (SMM), but can discovery descriptive behaviors that can be easily sampled. Furthermore, we show that through purely reward-free optimization, Skill DT is still competitive with supervised offline RL approaches on the D4RL benchmark.

## 1 INTRODUCTION

Reinforcement Learning (RL) has been incredibly effective in a variety of online scenarios such as games and continuous control environments (Li, 2017). However, they generally suffer from sample inefficiency, where millions of interactions with an environment are required. In addition, efficient exploration is needed to avoid local minimas (Pathak et al., 2017; Campos et al., 2020). Because of these limitations, there is interest in methods that can learn diverse and useful primitives without supervision, enabling better exploration and re-usability of learned skills (Eysenbach et al., 2018; Strouse et al., 2021; Campos et al., 2020). However, these online skill discovery methods still require interactions with an environment, where access may be limited.

This requirement has sparked interest in Offline RL, where a dataset of trajectories is provided. Some of these datasets (Fu et al., 2020) are composed of large and diverse trajectories of varying performance, making it non trivial to actually make proper use of these datasets; simply applying behavioral cloning (BC) leads to sub-optimal performance. Recently, approaches such as the Decision Transformer (DT) (Chen et al., 2021) and the Trajectory Transformer (TT) (Janner et al., 2021), utilize Transformer architectures (Vaswani et al., 2017) to achieve high performance on Offline RL benchmarks. Furuta et al. (2021) showed that these methods are effectively doing hindsight information matching (HIM), where the policies are trained to estimate a trajectory that matches given target statistics of future information. The work also generalizes DT as an information-statistic conditioned policy, Generalized Decision Transformer (GDT). This results in policies with different capabilities, such as supervised learning and State Marginal Matching (SMM) (Lee et al., 2019), just by simply varying different information statistics.

In the work presented here, we take inspiration from the previously mentioned skill discovery methods and introduce *Skill Decision Transformers* (Skill DT), a special case of GDT, where we wish to condition action predictions on skill embeddings and also *future* skill distributions. We show that Skill DT is not only able to discovery a number of discrete behaviors, but it is also able to effectively match target trajectory distributions. Furthermore, we empirically show that through pure unsupervised skill discovery, Skill DT is actually able to discover high performing behaviors that match or achieve higher performance on D4RL benchmarks (Fu et al., 2020) compared to other state-of-the-art offline RL approaches.

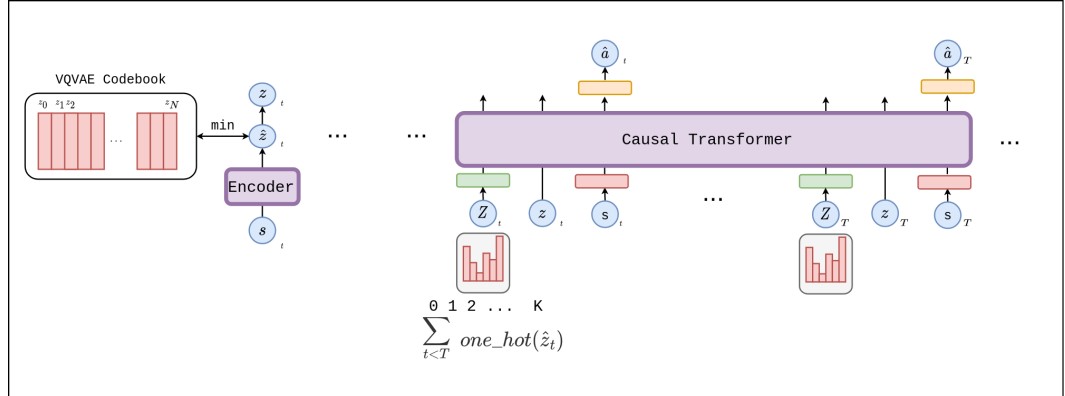

Figure 1: Skill Decision Transformer. States are encoded and clustered via VQ-VAE codebook embeddings. A Causal Transformer, similar to the original DT architecture, takes in a sequence of states, a latent skill distribution, represented as the normalized summed future counts of VQVAE encoding indices (details can be found in the "generate_histogram" function in A.5), and the corresponding skill encoding of the state at timestep $t$.

Our method predicts actions, conditioned by previous states, skills, and distributions of future skills. Empirically, we show that Skill DT can not only perform SMM on target trajectories, but can also match or achieve higher performance on D4RL benchmarks (Fu et al., 2020) compared to other state-of-the-art offline RL approaches. Skill DT also has the added benefit of using discrete skills, which are useful for easily sampling diverse behaviors.

## 2    RELATED WORK

### 2.1    SKILL DISCOVERY

Many skill methods attempt to learn a latent skill conditioned policy $\pi(a|s, z)$, where state $s \sim p(s)$ and skill $z \sim Z$, that maximizes mutual information between $S$ and $Z$ (Gregor et al., 2016; Sharma et al., 2019; Eysenbach et al., 2018). Another way of learning meaningful skills is through variational inference, where $z$ is learned via a reconstruction loss (Campos et al., 2020). Explore, Discover and Learn (EDL) (Campos et al., 2020) is an approach, which discovers a discrete set of skills by encoding states via a VQ-VAE: $p(z|s)$, and reconstructing them: $p(s|z)$. We use a similar approach, but instead of reconstructing states, we utilize offline trajectories and optimize action reconstruction directly ($p(a|s, z)$). Since our policy is autoregressive, our skill encoding actually takes into account temporal information, leading to more descriptive skill embeddings. Offline Primitive Discovery for Accelerating Offline Reinforcement Learning (OPAL) (Ajay et al., 2020), also discovers offline skills temporally, but instead uses a continuous distribution of skills. These continuous skills are then sample by a hierarchical policy that is optimized by task rewards. Because our approach is completely supervised, we wish to easily sample skills. To simplify this, we opt to use a discrete distribution of skills. This makes it trivial to query the highest performing behaviors, accomplished by just iterating through the discrete skills.

### 2.2    STATE MARGINAL MATCHING

State marginal matching (SMM) (Lee et al., 2019) involves finding policies that minimize the distance between the marginal state distribution that the policy represents $p^\pi(s)$, and a target distribution $p^*(s)$. These objectives have an advantage over traditional RL objectives in that they do not require any rewards and are guided towards exploration (Campos et al., 2020). CDT has shown impressive SMM capabilities by utilizing binned target state distributions to condition actions in order to match the given target state distributions. However, using CDT in a real environment is difficult because target distributions must be provided, while Skill DT learns discrete skills that can be sampled easily. Also, CDT requires a low dimensional state space, while Skill DT in theory can work on any type of input as long as it can be encoded effectively into a vector.

## 3 PRELIMINARIES

In this work, we consider learning in environments modelled as Markov decision processes (MDPs), which can be described using varibles $(S, A, P, R)$, where $S$ represents the state space, $A$ represents the action space, and $P(s_{t+1}|s_t, a_t)$ represents state transition dynamics of the environment.

### 3.1 GENERALIZED DECISION TRANSFORMER

The Decision Transformer (DT) (Chen et al., 2021) represents RL as a sequence modelling problem and uses a GPT architecture Alec Radford & Sutskever (2018) to predict actions autoregressively. Specifically, DT takes in a sequence of RTGs, states, and actions, where $R_t = \sum_t^T r_t$, and trajectory $\tau = (R_0, s_0, a_0, ..., R_{|\tau|}, s_{|\tau|}, a_{|\tau|})$. DT uses $K$ previous tokens to predict $a_t$ with a deterministic policy which is optimized by a mean squared error loss between target and predicted actions. For evaluation, a target return $\hat{R}_{target}$ is provided and DT attempts to achieve the targeted return in the actual environment. Furuta et al. (2021) introduced a generalized version of DT, Generalized Decision Transformer (GDT). GDT provides a simple interface for representing a variety of different objectives, configurable by different information statistics (for consistency, we represent variations of GDT with $\pi^{gdt}$):

$$\tau_t = s_t, a_t, r_t, ..., s_T, a_T, r_T, I^\phi = \text{information statistics function}$$

Generalized Decision Transformer (GDT):

$$\pi^{gdt}(a_t|I^\phi(\tau_0), s_0, a_0..., I^\phi(\tau_t), s_{t-1}, a_{t-1})$$

Decision Transformer (DT):

$$\pi_{dt}^{gdt}(a_t|I_{dt}^\phi(\tau_0), s_0, a_0, ..., I_{dt}^\phi(\tau_t), s_{t-1}, a_{t-1}), \text{ where } I_{dt}^\phi(\tau_t) = \sum_t^T \gamma * r_t, \gamma = \text{discount factor}$$

Categorical Decision Transformer (CDT):

$$\pi_{cdt}^{gdt}(a_t|I_{cdt}^\phi(\tau_0), s_0, a_0, ..., I_{cdt}^\phi(\tau_t), s_t, a_t), \text{ where } I_{cdt}^\phi(\tau_t) = histogram(s_t, ..., s_T)$$

CDT is the most similar to Skill DT – CDT captures future trajectory information using future state distributions, represented as histograms for each state dimension, essentially binning and counting the bin ids for each state dimension. Skill DT instead utilizes learned skill embeddings to generate future skill distributions, represented as histograms of **full** embeddings. In addition, Skill DT also makes use of the representation learnt by the skill embedding by also using it in tandem with the skill distributions.

## 4 SKILL DECISION TRANSFORMER

### 4.1 FORMULATION

Our Skill DT architecture is very similar to the original Decision Transformer presented in Chen et al. (2021). While the classic DT uses summed future returns to condition trajectories, we instead make use of learned skill embeddings and future *skill distributions*, represented as a histogram of skill embedding indices, similar to the way Categorical Decision Transformer (CDT) (Furuta et al., 2021) utilizes future state counts. One notable difference Skill DT has to the original Decision Transformer (Chen et al., 2021) and the GDT (Furuta et al., 2021) variant is that we omit actions in predictions. This is because we are interested in SMM through skills, where we want to extract as much information from states.

Formally, Skill DT represents a policy:

$$\pi(a_t|Z_{t-K}, z_{t-K}, s_{t-K}, ...Z_{t-1}, z_{t-1}, s_{t-1}),$$

where $K$ is the context length, and $\theta$ are the learnable parameters of the model. States are encoded as skill embeddings $\hat{z}_t$, which are then quantized using a learned codebook of embeddings $z =$

$argmin_n||\hat{z} - z_n||_2^2$. The future skill distributions are represented as the normalized histogram of summed future one hot encoded skill indices: $Z_t = \sum_t^T one\_hot(z_t)$. Connecting this to GDT, our policy can be viewed as:

$$\pi_{skill}^{gdt}(a_t|I_{skill}^{\phi}(\tau_0), s_0, ..., I_{skill}^{\phi}(\tau_t), s_t), \text{ where } I_{skill}^{\phi}(\tau_t) = (histogram(z_t, ..., z_T), z_t).$$

### 4.1.1 HINDSIGHT SKILL RE-LABELLING

Hindsight experience replay (HER) is a method that has been effective in improving sample-efficiency of goal-oriented agents (Andrychowicz et al., 2017; Rauber et al., 2017). The core concept revolves around *goal relabelling*, where trajectory goals are replaced by achieved goals vs. inteded goals. This concept of re-labelling information has been utilized in a number of works (Ghosh et al., 2019; Zheng et al., 2022; Faccio et al., 2022), to iteratively learn an condition predictions on target statistics. Bi-Directional Decision Transformer (BDT) (Furuta et al., 2021), utilizes an anti-causal transformer to encode trajectory information, and passes it into a causal transformer action predictor. At every training iteration, BDT re-labels trajectory information with the anti-causal transformer. Similarly, Skill DT re-labels future skill distributions at every training iteration. Because the skill encoder is being updated consistently and skill representations change during training, the re-labelling of skill distributions is required to ensure stability in action predictions.

### 4.2 ARCHITECTURE

**VQ-VAE Skill Encoder**. Many previous works have represented discrete skills as categorical variables, sampled from a categorical distribution prior (Strouse et al., 2021; Eysenbach et al., 2018). VQ-VAEs (van den Oord et al., 2017) have shown impressive capabilities with discrete variational inference in the space of computer vision (Razavi et al., 2019; Esser et al., 2020), planning (Ozair et al., 2021), and online skill discovery (Campos et al., 2020). Because of this, we use a VQ-VAE to quantize encoded states into a set of continuous skill embeddings. We encode states into vectors $z$, and quantize to nearest skill embeddings $\hat{z}$. To ensure stability, we minimize the loss:

$$VQLOSS(z, \hat{z}) = MSE(z, \hat{z}) \tag{1}$$

Where $\hat{z}$ is the output of the MLP encoder and $z$ is the nearest embedding in the VQ-VAE codebook.

Optimizing this loss minimizes the distance of our skill encodings with their corresponding nearest VQ-VAE embeddings. This is analagous to clustering, where we are trying to minimize the distance between datapoints and their actual cluster centers. In practice, we optimize this loss using an exponential moving average, as detailed in Lai et al. (2022).

**Causal Transformer**. The Causal Transformer portion of Skill DT shares a similar architecture to that of the original DT (Chen et al., 2021), utilizing a GPT (Alec Radford & Sutskever, 2018) model. It takes in input the last $K$ states $s_{t-K:t}$, skill encodings $z_{t-K:t}$, and future skill embedding distributions $Z_{t-K:t}$. As mentioned above, the future skill embedding distributions are calculated by generating a histogram of skill indices from timestep $t : T$, and normalizing them so that they add up to 1. For states and skill embedding distributions, we use learned linear layers to create token embeddings. To capture temporal information, we also learn a timestep embedding that is added to each token. Note that we don't tokenize our skill embeddings because we want to ensure that we don't lose important skill embedding information. It's important to note that even though we don't add timestep embeddings to the skill embeddings, they still capture temporal behavior because the attention mechanism (Vaswani et al., 2017) of the causal transformer attends the embeddings to temporally conditioned states and skill embedding distributions. The VQ-VAE and Causal Transformer components are shown visually in Fig. 1.

### 4.3 TRAINING PROCEDURE

Training Skill DT is very similar to how other variants of GDT are trained (CDT, BDT, DT, etc.). First, before every training iteration we re-label skill distributions for every trajectory using our VQ-VAE encoder. Afterwards, we sample minibatches of sequence length $K$, where timesteps are sampled uniformly. Specifically, at every training iteration, we sample $\tau = (s_t, ...s_{t+K}, a_t, ...a_{t+K})$,

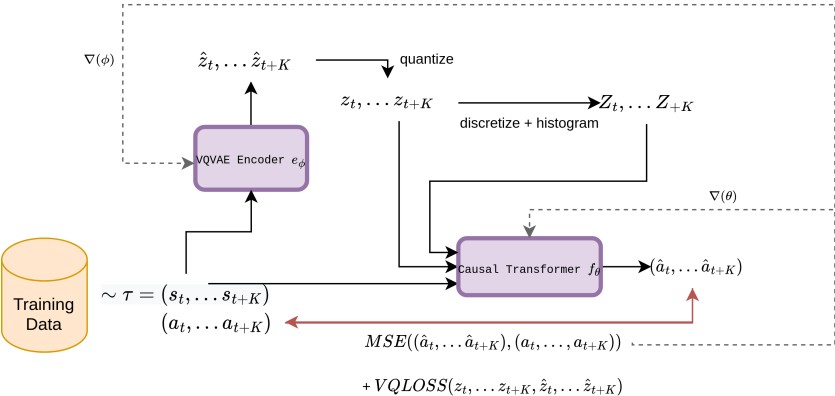

Figure 2: Training procedure for Skill Decision Transformer. sub-trajectories of states of length $k$ are sampled from the dataset and encoded into in latents and discretized. All three variables are passed into the causal transformer to output actions. The VQVAE parameters and Causal Transformer parameters are backpropagated directly using an MSE loss and VQ-VAE regularization loss, shown in 1

where $t$ is sampled uniformly for each trajectory in the batch. The sampled states, $(s_t, ...s_{t+K})$, are encoded into skill embeddings using the VQVAE encoder. We then pass in the states, encoded skills, and skill distributions into the causal transformer to output actions. Like the original DT (Chen et al., 2021), we also did not find it useful to predict states or skill distributions, but it could be useful for actively predicting skill distributions without having to actually provide states to encode. This is a topic we hope to explore more in the future. The VQVAE encoder and causal transformer are updated by backpropagation through an MSE loss between target actions and predicted actions and the VQVAE regularization loss referenced in 1. The simplified training procedure is shown in Algorithm 1.

---

**Algorithm 1** Offline Skill Discovery with Skill Decision Transformer

---

**Initialize** offline dataset $D$, Causal Transformer $f_\theta$, VQVAE Encoder $e_\phi$, context length $K$, num updates per iteration $J$

**for** training iterations $i = 1...N$ **do**

    Sample timesteps uniformly: $t \in 1, ...max\_len$

    Label dataset trajectories with skill distributions $Z_{\tau_t} = \sum_t^T one\_hot(z_t)$ for all $t, ..|\tau|$

    Sample batch of trajectory states: $\tau = (s_t, ...s_{t+K}, a_t, ...a_{t+K})$

    **for** j = 1...$J$ **do**

        $\hat{z}_{\tau_{t:t+K}} = (e_\phi(s_t), ...e_\phi(s_{t+K}))$ Encode skills

        $z_{\tau_{t:t+K}} = quantize(\hat{z}_{\tau_{t:t+K}})$ Quantize skills with VQVAE

        $\hat{a}_{\tau_{t:t+K}} = f_\theta(Z_{\tau_t}, z_{\tau_t}, s_t, ..., Z_{\tau_{t+K}}, z_{\tau_{t+K}}, s_{t+K})$

        $L_{\theta,\phi} = \frac{1}{K} \sum_t^{t+K} (a_t - \hat{a}_t)^2 + VQLOSS_\phi(z_{\tau_{t:t+K}}, \hat{z}_{\tau_{t:t+K}})$

        backprop $L_{\theta,\phi}$ w.r.t $\theta, \phi$

    **end for**

**end for**

---

## 5 EXPERIMENTS

### 5.1 TASKS AND DATASETS

For evaluating the performance of Skill DT, we use tasks and datasets from the D4RL benchmark (Fu et al., 2020). D4RL has been used as a standard for evaluating many offline RL methods (Kumar

et al., 2020; Chen et al., 2021; Kostrikov et al., 2021; Zheng et al., 2022). We evaluate our methods on mujoco gym continuous control tasks, as well as two antmaze tasks. Images of some of these environments can be seen in A.4

## 5.2 EVALUATING SUPERVISED RETURN

**Can Skill DT achieve near or competitive performance, using only trajectory information, compared to supervised offline RL approaches?**

| Mujoco Mean Results | | | | | | | | |
|---|---|---|---|---|---|---|---|---|
| Env Name | DT | CQL | IQL | OPAL | KMeans DT | **Skill DT (best skill)** | num skills (Skill DT) | Dataset Max Reward |
| walker2d-medium | 74 | 79 | 78.3 | — | 76 | **82** | 10 | 92 |
| halfcheetah-medium | 43 | 44 | **47** | — | 43 | 44 | 10 | 45 |
| ant-medium | 94 | — | 101 | — | 100 | **106** | 10 | 107 |
| hopper-medium | 68 | 58 | 66 | — | 66 | **76** | 32 | 100 |
| halfcheetah-medium-replay | 37 | **46** | 44 | — | 39 | 41 | 32 | 42 |
| hopper-medium-replay | 63 | **95** | **95** | — | 71 | 81 | 32 | 99 |
| antmaze-umaze | 59 | 75 | 88 | — | 73 | **100** | 32 | 100 |
| antmaze-umaze-diverse | 53 | 84 | 62 | — | 67 | **100** | 32 | 100 |
| antmaze-medium-diverse | 0 | 61 | **71** | — | 0 | 13 | 64 | 100 |
| antmaze-medium-play | 0 | 54 | 70 | **81** | 0 | 0 | 64 | 100 |

Table 1: Average normalized returns on Gym and AntMaze tasks. We obtain some results as reported on other works (Chen et al., 2021; Kumar et al., 2020; Kostrikov et al., 2021; Zheng et al., 2022), and calculate Skill DT's returns as an average over 4 seeds (for gym) and 15 (for antmaze). Skill DT outperforms the baselines on most tasks, but fails to beat them on replay tasks and antmaze-medium. However, Skill DT can consistently solve the antmaze-umaze tasks.

Other offline skill discovery algorithms optimize hierarchical policies via supervised RL, utilizing the learned primitives to maximize rewards of downstream tasks (Ajay et al., 2020). However, because we are interested in evaluating Skill DT **without** rewards, we have to rely on learning enough skills such that high performing trajectories are represented. To evaluate this in practice, we run rollouts for each unique skill and take the maximum reward achieved. Detailed python sudocode for this is provided in A.5. For a close skill-based comparison to Skill DT, we use a K-Means augmented Decision Transformer (K-Means DT). K-Means DT differs from Skill DT in that instead of learning skill embeddings, instead we cluster states via K-Means and utilize the cluster centers as the skill embeddings.

Surprisingly, through just pure unsupervised skill discovery, we are able to achieve competitive results on Mujoco continuous control environments compared to state-of-the-art offline reinforcement learning algorithms (Kumar et al., 2020; Kostrikov et al., 2021; Chen et al., 2021). As we can see in our results in Table 1, Skill DT outperforms other baselines on most of the tasks and DT on all of the tasks. However, it performs worse than the other baselines on the antmaze-medium / -replay tasks. We hypothesize that Skill DT performs worse in these tasks because they contain multimodal and diverse behaviors. We think that with additional return context or online play, Skill DT may be able to perform better in these environments, and we hope to explore this as future work. Skill DT, like the original Decision Transformer (Chen et al., 2021), also struggles on harder exploration problems like the antmaze-medium environments. Methods that perform well on these tasks usually utilize dynamic programming like Trajectory Transformer(Janner et al., 2021) or hierarchical reinforcement learning like OPAL (Ajay et al., 2020). Even though Skill DT performs marginally better than DT, there is still a lot of room for improvement in future work.

## 6 DISCUSSION

### 6.1 ABLATION STUDY

**What is the effect of the number of skills?**

| Ablation Results | | | | |
|---|---|---|---|---|
| Env Name | 5 skills | 10 skills | 16 skills | 32 skills |
| walker2d-medium | 80 | 82 | 82 | 82 |
| halfcheetah-medium | 44 | 44 | 44 | 44 |
| ant-medium | 100 | 106 | 106 | 106 |
| hopper-medium | 65 | 70 | 76 | 76 |
| hopper-medium-replay | 28 | 31 | 46 | 81 |
| halfcheetah-medium-replay | 34 | 39 | 41 | 41 |
| ant-umaze | 80 | 100 | 100 | 100 |
| ant-umaze-diverse | 66 | 100 | 100 | 100 |

Table 2: Best reward obtained from skills for a varying number of skills

Because Skill DT is a completely unsupervised algorithm, evaluating supervised return requires evaluating every learnt skill and taking the one that achieves the maximum reward. This means we are relying entirely on Skill DT's ability to capture behaviors from high performing trajectories from the offline dataset. We found that increasing the number of skills has less of an effect on performance, in environments that have a large number of successful trajectories (-medium environments). We hypothesize that these datasets have unimodal behaviors, and Skill DT does not need many skills to capture descriptive information from the dataset. However, for multimodal datasets (such as the -replay environments), Skill DT's performance improves with an increasing number of skills. In general, using a larger number of skills can help performance, but the tradeoff is increased computation time because each skill needs to be evaluated in the environment. These results are reported in Table 2. Images of skills learnt can be seen in A.4

### 6.2 SMM WITH LEARNED SKILLS

**How well can Skill DT reconstruct target trajectories and perform SMM in a zero shot manner?**

Ideally, if an algorithm is effective at SMM, it should be able to reconstruct a target trajectory in an actual environment. That is, given a target trajectory, the algorithm should be able to rollout a similar trajectory. The original DT can actually perform SMM well, on **offline** trajectories. However, when actually attempting this in an actual environment, it is unable to reconstruct a target trajectory because it is unable to be conditioned on accurate future state trajectory information. Skill DT, similar to CDT, is able to perform SMM in an actual environment because it encodes future state information into skill embedding histograms. The practical process for this is fairly simple and detailed in Algorithm 2. In addition to state trajectories, learned skill distributions of the reconstructed trajectory and the target trajectory should also be close. We investigate this by looking into target trajectories from antmaze-umaze-v2, antmaze-umaze-diverse-v2. For a more challenging example, we handpicked a trajectory from antmaze-umaze-diverse that is unique in that it has a loop. Even though the trajectory is unique, Skill DT is still able to roughly recreate it in a zero shot manner (Fig. 3), with rollouts also including a loop in the trajectory. Additional results can be found in A.3.

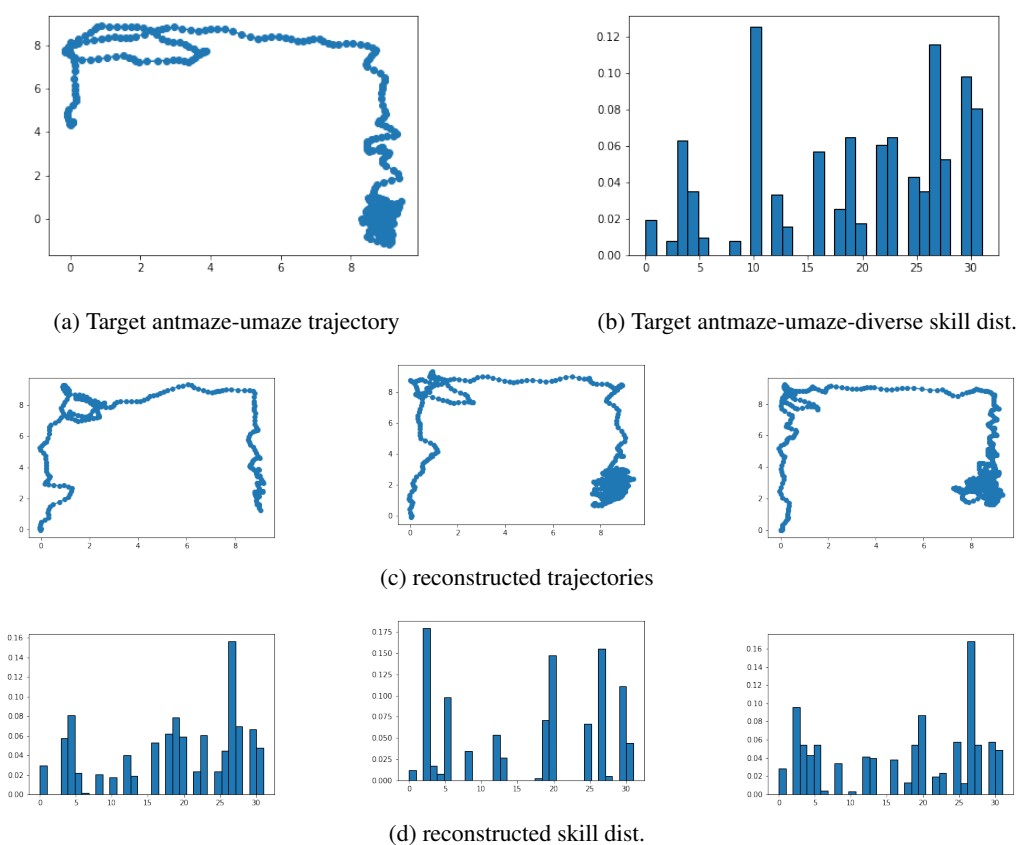

(a) Target antmaze-umaze trajectory

(b) Target antmaze-umaze-diverse skill dist.

(c) reconstructed trajectories

(d) reconstructed skill dist.

Figure 3: From the antmaze-umaze-diverse Environment: The target trajecty is complex, with a loop and with noisy movement. Reconstructed rollouts also contain a loop

## 6.3 SKILL DIVERSITY AND DESCRIPTIVENESS

**How diverse and descriptive are the skills that Skill DT discovers?**.

In order to evaluate Skill DT as a skill discovery method, we must show that behaviors are not only diverse but are descriptive, or more intuitively, **distinguishable**. We are able to visualize the diversity of learned behaviors by plotting each trajectory generated by a skill on both antmaze-umaze and ant environments, shown below. To visualize Skill DT's ability to describe states, we show the the projected skill embeddings and quantized skill embedding clusters (Fig. 5). For a diversity metric, we utilize a Wasserstein Distance metric between skill distributions (normalized between [0, 1]), similar to the method proposed in (Furuta et al., 2021). We report this metric in Table 3.

| Wasserstein Distances | | | |
|---|---|---|---|
| Env Name | min | max | avg |
| walker2d-medium | 0.007 | 0.015 | 0.010 |
| ant-medium | 0.007 | 0.012 | 0.008 |
| hopper-medium-replay | 0.009 | 0.036 | 0.027 |
| halfcheetah-medium-replay | 0.011 | 0.033 | 0.019 |
| ant-umaze-diverse | 0.008 | 0.026 | 0.011 |

Table 3: Wasserstein distance metric (computed between each skill and all others). In tasks with unimodal behaviors (-medium), Skill DT discovers skills that result in trajectories that are more similar to eachother than more complex tasks (-replay and antmaze)

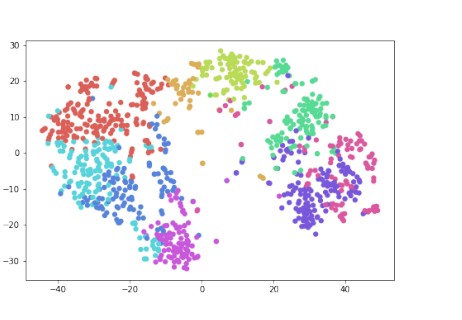
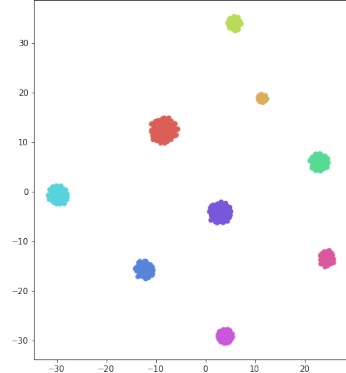

Figure 5: t-SNE projections of Ant-v2 states. Left: States are encoded into unquantized skill embeddings and projected via TSNE. Right: States are encoded into quantized skill embeddings and projected via TSNE

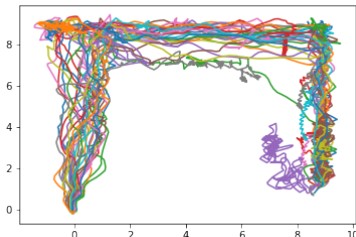
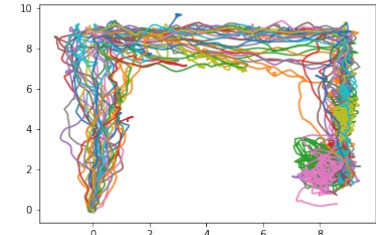

Figure 4: Left: ant-umaze-v2, Right: ant-umaze-diverse-v2. Trajectories made using 32 skills for both ant-umaze variants. The diverse variant contains lots of noisy trajectories, but Skill DT is still learn diverse and distinguishable skills

### 6.4 LIMITATIONS AND FUTURE WORK

Our approach is powerful because it is unsupervised, but it is also limited because of it. Because we do not have access to rewards, we rely on pure offline diversity to ensure that high performing trajectories are learned and encoded into skills that can be sampled. However, this is not very effective for tasks that require dynamic programming or longer sequence prediction. Skill DT could benefit from borrowing concepts from hierarchical skill discovery (Ajay et al., 2020) to re-use learned skills on downstream tasks by using an additional return-conditioned model. In addition, it would be interesting to explore an online component to the training procedure, similar to the work in Zheng et al. (2022).

## 7 CONCLUSION

We proposed Skill DT, a variant of Generalized DT, to explore the capabilities of offline skill discovery with sequence modelling. We showed that a combination of LLMs, hindsight-relabelling can be incredibly useful for extracting information from diverse offline trajectories.. On standard offline RL environments, we showed that Skill DT is capable of learning a rich set of behaviors and can perform zero-shot SMM through state-encoded skill embeddings. Skill DT can further be improved by adding an online component, a hierarchical component that utilizes returns, and improved exploration.

ACKNOWLEDGEMENTS

Removed for blind review.

REPRODUCIBILITY STATEMENT

The code to to reproduce every experiment in this paper is available at [removed].

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

# A APPENDIX

## A.1 DATASET STATISTICS

| Dataset Stats | | | | | | | | | |
|---|---|---|---|---|---|---|---|---|---|
| Env Name | state dim | act dim | num trajec- tories | avg dataset reward | max dataset reward | min dataset reward | avg dataset d4rl | max dataset d4rl | min dataset d4rl |
| walker2d - medium | 17 | 6 | 1190 | 2852 | 4227 | -7 | 62 | 92 | 0 |
| halfcheetah -medium | 17 | 6 | 1000 | 4770 | 5309 | -310 | 41 | 45 | 0 |
| ant - medium | 111 | 8 | 1202 | 3051 | 4187 | -530 | 80 | 107 | -5 |
| hopper -medium | 11 | 3 | 2186 | 1422 | 3222 | 316 | 44 | 100 | 10 |
| halfcheetah -medium -replay | 17 | 6 | 202 | 3093 | 4985 | -638 | 27 | 42 | -3 |
| hopper- medium -replay | 11 | 3 | 1801 | 529 | 3193 | -0.5 | 17 | 99 | 1 |
| antmaze- umaze | 29 | 8 | 2815 | 0.5 | 1 | 0 | 52 | 100 | 0 |
| antmaze- umaze -diverse | 29 | 8 | 1011 | 0.012 | 1 | 0 | 1.2 | 100 | 0 |
| antmaze- medium -diverse | 29 | 8 | 1137 | 0.125 | 1 | 0 | 12.5 | 100 | 0 |
| antmaze- medium -play | 29 | 8 | 1204 | 0.2 | 1 | 0 | 20.0 | 100 | 0 |

Table 4: Dataset statistics

## A.2 HYPERPARAMETERS

| Common Hyper Parameters for Causal Transformer | |
|---|---|
| hyperparameter | value |
| Number of layers | 4 |
| Number of attention heads | 4 |
| Embedding dimension | 256 |
| Context Length | 20 |
| Dropout | 0.0 |
| Batch Size | 256 |
| Updates between rollouts | 50 |
| lr | 1e-4 |
| gradient norm | 0.25 |

Table 5

## A.3 FEW SHOT TARGET TRAJECTORY RECONSTRUCTION

Skill DT, like other skill discovery methods, can use its state-to-skill encoder to guide its actions towards a particular goal. In this case, we are interested in recreating target trajectories as close as possible. The detailed algorithm for few shot skill reconstruction: 5

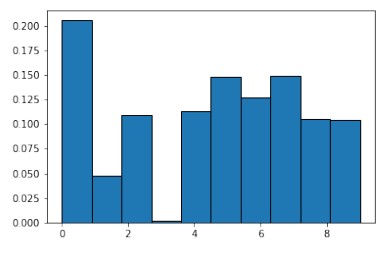

(a) Target Ant Skill Distribution

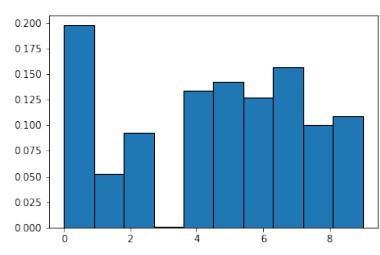

(b) Reconstructed Ant Skill Distribution

Figure 7: From the Ant-v2 Environment: Skill distributions of a target trajectory and the reconstructed trajectory from rolling out in the environment. Because Ant-v2 is a simpler environment, we can see that the reconstructed skill distributions are very close to the target.

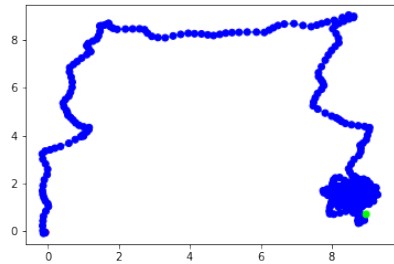

(a) Target antmaze-umaze x-y trajectory

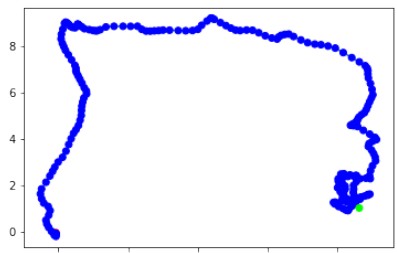

(b) Reconstructed antmaze-umaze x-y trajectory

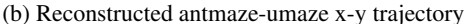

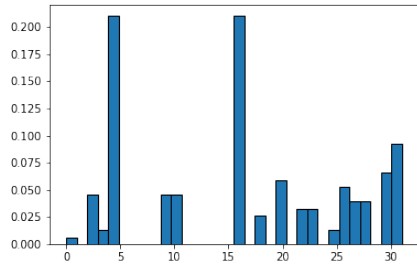

(c) Target antmaze-umaze skill dist.

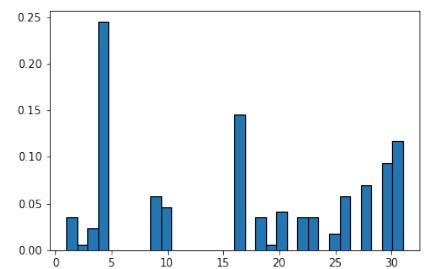

(d) Reconstructed antmaze-umaze skill dist.

Figure 6: From the Antmaze-Umaze Envidiverseronment: One of the longer and highest performing trajectories in the dataset is reconstructed by Skill DT. The trajectory is not quite identical to the target, but it follows a similar path, where it hugs the edges of the maze just like the target.

---

**Algorithm 2** Reconstructing target trajectories

---

**Initialize** target trajectory $\tau$, skill encoder $E_\phi$, Skill DT transformer $\pi$

1. $(s_0^{target}, ..., s_T^{target}) \leftarrow \tau$,                  ▷ Extract states from target trajectory

2. $(z_0, ..., z_T), (zindex_0, ..., zindex_T) = E_\phi(s_0^{target}, ..., s_T^{target})$,        ▷ encode states

3. $(Z_0, ..., Z_T) = histogram(zindex_0, ..., zindex_T)$, ▷ Create skill distributions by creating a histogram of skill encoding indices

4. $\sim \pi(a|Z_0, z_0, s_0, ...)$,                    ▷ rollout in real environment, see A.5 for details

---

## A.4 Trajectory Skill Visualizations

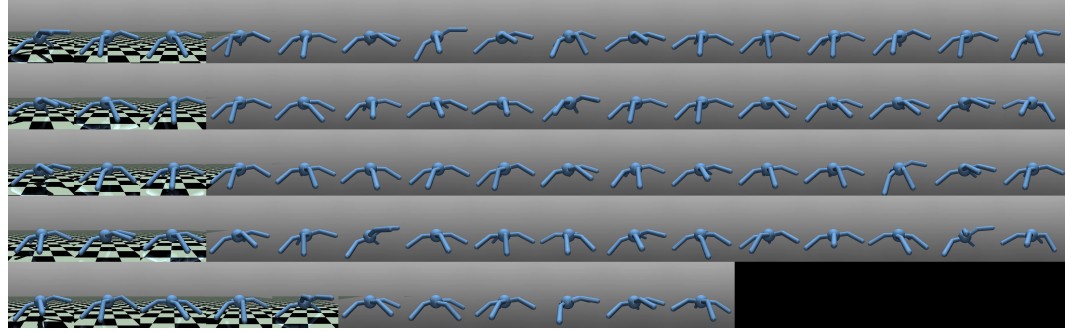

Figure 8: Skills learned in the ant-medium-v2 environment. Each row corresponds to a skill

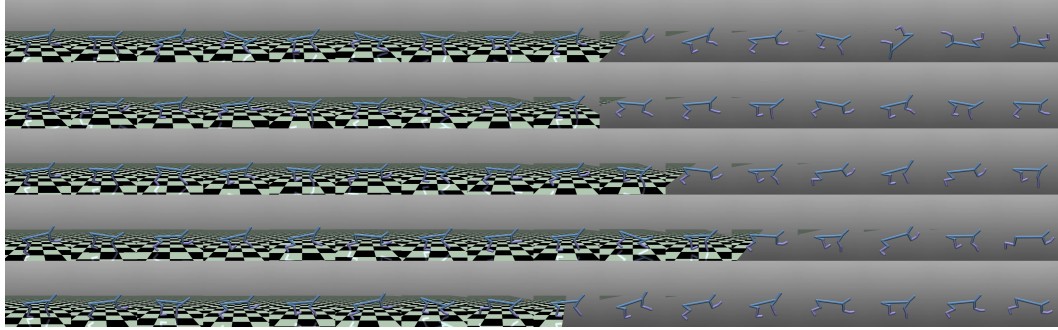

Figure 9: Skills learned in the halfcheetah-medium-replay-v2 environment. Each row corresponds to a skill

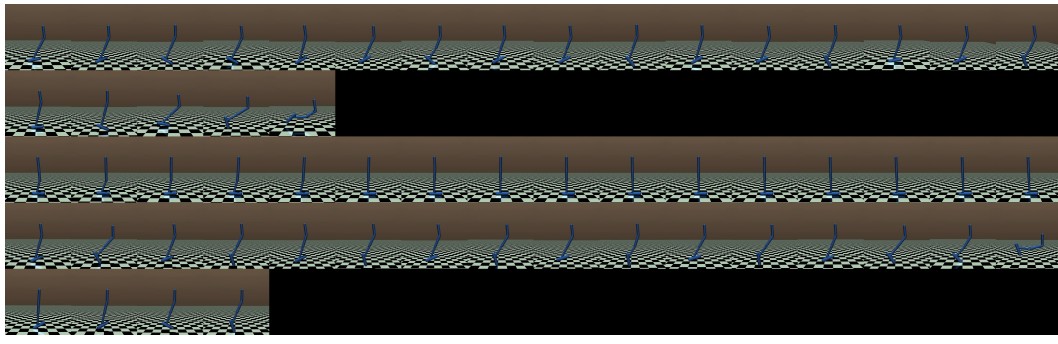

Figure 10: Skills learned in the hopper-medium-replay-v2 environment. Each row corresponds to a skill

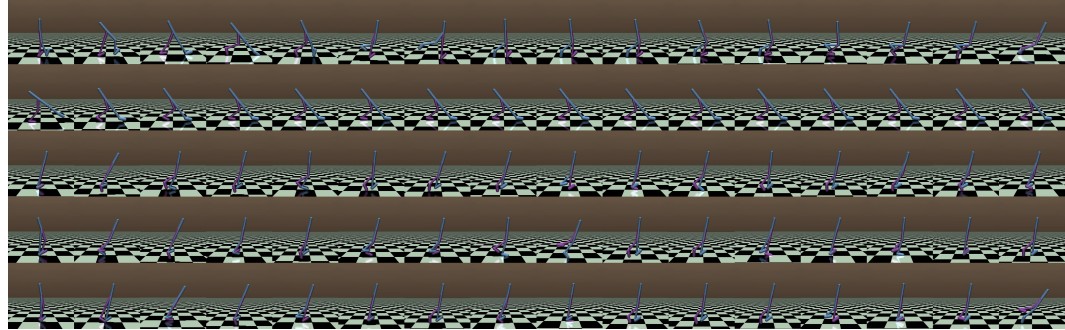

Figure 11: Skills learned in the walker-medium-v2 environment. Each row corresponds to a skill

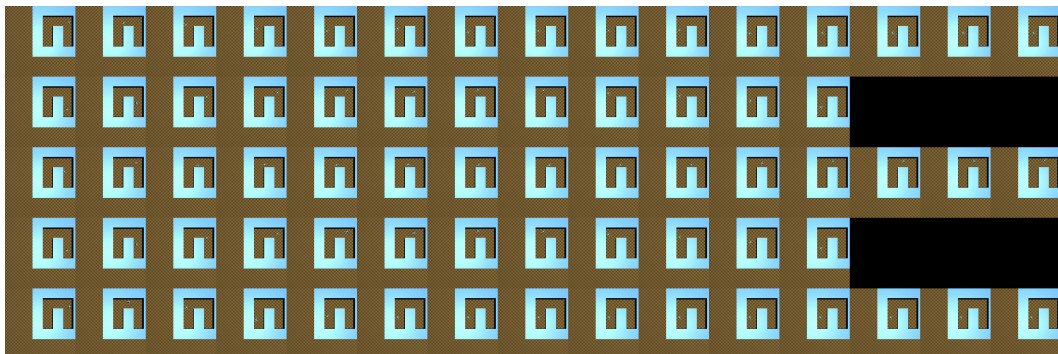

Figure 12: Skills learned in the ant-umaze-diverse-v2 environment. Each row corresponds to a skill

### A.5   EVALUATING SKILL DT'S PERFORMANCE

Because Skill DT is a purely unsupervised algorithm, to evaluate the performance in an actual environment, we perform rollouts for each skill and evaluate each to determine which is the best. To do this, we first populate a buffer of skills (here we denote with $z$) and skill histograms $Z$. When we rollout in the actual environment, the causal transformer utilizes this buffer to actually make predictions. However, it updates the skill encodings that it **actually** sees in the environment at each timestep. This is because even though the policy is completely conditioned to follow a single skill, it may end up reaching states that are classified under another. Python sudocode shown below:

```python
def generate_histogram(one_hot_skill_ids):
    trajectory_length = len(one_hot_skill_ids)
    histogram = torch.tensor(copy(one_hot_skill_ids))

    # reverse order
    for i in range(trajectory_length-1, -1, -1):
        if i != trajectory_length - 1
            histogram[i] = histogram[i] + histogram[i+1]
    return histogram / histogram.sum(-1) # normalize in range [0, 1]

def evaluate_skill_dt(skill_dt, env, max_steps, context_len):
    num_skills = skill_dt.num_skills
    rewards = []
    for skill_id in range(num_skills):
        skill_ids = repeat(skill_id, max_steps)
        # create on_hot skill ids
        # ex: one_hot([1,1], 5) = [[0, 1, 0, 0, 0],[0, 1, 0, 0, 0]]
        one_hot_skill_ids = one_hot(skill_ids, num_skills)

        # initialize_state
        state = env.reset()
        t = 0
        total_reward = 0

        state_buffer = zeros(max_steps)
        z_buffer = zeros(max_steps)

        while t < max_steps:
            # z is vqvae embedding
            # skill_id is the index of the vqvae embedding in codebook
            z, skill_id = skill_dt.encode_skill(state)
            one_hot_skill_ids[t] = one_hot(skill_id)
            Z = skill_dt.generate_histogram(one_hot_skill_ids) # create
                                                    histograms
            state_buffer[t] = state
            z_buffer[t] = z
            if t < context_len:
                curr_states = state_buffer[t:t+context_len]
                curr_z = z_buffer[t:t+context_len]
                curr_Z = Z[t:t+context_len]
                actions = skill_dt.causal_transformer(curr_Z, curr_z,
                                                      curr_states)
                action = actions[t]
            else:
                curr_states = state_buffer[t-context_len+1:t+1]
                curr_z = z_buffer[t-context_len+1:t+1]
                curr_Z = Z[t-context_len+1:t+1]
                actions = skill_dt.causal_transformer(curr_Z, curr_z,
                                                      curr_states)
                action = actions[-1]
            state, reward, done = env.step(action)
            total_reward += reward
```

```
            if done:
                break
        rewards.append(total_reward)
    return max(rewards)
```

