# OpenReview forum: "Skill Decision Transformer"
_ICLR.cc/2023/Conference — Submitted to ICLR 2023_

### Official Review · Reviewer_L7Nj · 2022-10-24

**Confidence:** 5
**Correctness:** 2
**Technical Novelty And Significance:** 2
**Empirical Novelty And Significance:** 2
**Recommendation:** 3

**Clarity, Quality, Novelty And Reproducibility:**

### Quality
There are some points that should be improved in writing.
- The title in the paper should be fixed (Skill Discovery Decision Transformer --> Skill Decision Transformer).
- In Section 3, the definition of $\tau_t$ is missing.
- The explanation of GDT in Section 3.1 seems to miss $a_t$.
- In Section 3.1, the information statistics of DT should be $\Sigma \gamma^t r_t$ rather than $\Sigma \gamma * r$ (\gamma is also not defined).
- Citation of "a GPT architecture ..." seems broken.
- In Section 5.1, "timeplapse" --> "timelapse".
- In Algorithm 1, $e_{\phi}(s_t), .. e_{\phi}(s_t)$ --> $e_{\phi}(s_t), .. e_{\phi}(s_{t+K})$.
- The GDT-like formalization of Skill DT in Section 4.1 might be wrong: $I = (histogram, z_t)$ --> $I = histogram$?
- The sentence in Section 7 has duplicate periods.

### Clarity
There are some unclear points in the paper.
- Compared to DT or GDT, Skill DT doesn't have actions as inputs. Is there any justification/explanation?
- How does skill DT prepare the future skill distribution in the evaluation time?
- The suffix of Figure 1 should reflect context length: i.e. $t - K, ..., t$ or $t, .., t+K$, instead of $t, ..., T$.
- Continuous skill-based methods often assume gaussian distribution as a skill distribution, which is easy to sample latent skills. It should be clarified how "easily sampling diverse behaviors" in Skill DT. Since Skill DT seems to sample the discrete skill variable at every timestep, it seems difficult to sample desired / consistent behaviors.
- Figure 3 said "Skill DT is still learn diverse and distinguishable skills", but I don't think those trajectories are distinguishable. They are very cluttered.
- The visualization in Figure 6 might not be so different from each other compared to Figure 5. Some quantitative metrics would be helpful.
- The number of skills in Figure 5 seems different between VQ-VAE and k-means. It should be aligned.
- The ablation of num_skills presented in Table 1 could be important. While the medium dataset has unimodal behaviors, it requires 10 skill variables, which doesn't seem intuitive.



### Originality
The quantization of offline behavioral data for sequential modeling seems to be shown in GDT paper. The originality of this paper is the replacement of the state-discretization / anti-causal transformer encoder with VQ-VAE encoder.



**Strength And Weaknesses:**

### Strength
- While the quantization of offline behavioral data for sequential modeling seems to be shown in GDT paper, the learned skill embedding with VQ-VAE may be a novel approach for transformer-based offline RL.
- The experimental results presented in Table 1 show a certain amount of improvement among competitive offline RL methods (DT, CQL, IQL).

### Weaknesses
 - The domain where Skill DT is effective seems limited. For instance, while Skill DT shows notable performance in antmaze-umaze or mujoco-medium settings that have unimodal behaviors, it doesn't show improvement in antmaze-medium or mujoco-medium-replay that have multimodal behaviors. Since skill discovery methods usually learn diverse and multimodal behaviors, these trends are not intuitive.

- There is no skill-based baseline. GDT-variants (CDT/BDT), k-means clustering-conditioned DT, or some offline skill-based method (OPAL, LiSP [1]) can be relevant baselines.

[1] https://arxiv.org/abs/2012.03548

**Summary Of The Paper:**

This paper proposes Skill Decision Transformer (Skill DT), a novel offline RL method with transformer-based sequential modeling, which can be interpreted as an extension of Generalized Decision Transformer, proposed by Furuta et al.
Skill DT first encodes each state into a discrete latent variable $z_t$ with VQ-VAE codebook, converts it into a one-hot vector, and then makes a histogram over the future time steps. After that, causal transformer takes the sequence of such future-aggregated skill distribution, skill, and state as an input, and auto-regressively predicts the actions at each time step.
The experiments show that Skill DT achieves superior or comparable performance to other existing offline RL methods (DT, CQL, IQL) with several D4RL MuJoCo-locomotion, and antmaze datasets.

**Summary Of The Review:**

Combining VQ-VAE with transformer-based offline RL (DT, GDT) might be an important direction to leverage diverse and unstructured behavioral datasets. However, this paper doesn't have enough experimental evaluations and has many unclear points that should be revised. Considering those aspects, I vote for rejection.

---

> ### Author Response · Authors · 2022-11-19
> **Response to Reviewer L7Nj**
>
> Thank you reviewer L7Nj for the great feedback! We'd like to address some of your comments:
>
> Thank you for pointing out the discrepancies in our formulas! We have made some edits and hopefully covered all the great points you've made.
>
> ***"in antmaze-medium or mujoco-medium-replay that have multimodal behaviors. Since skill discovery methods usually learn diverse and multimodal behaviors, these trends are not intuitive."***
>
> As shown in the ablation studies, Skill DT does perform better in these tasks with a larger number of skills. And we observe that while Skill DT performs worse than online methods, it outperforms offline methods DT and K Mean DT.
>
> ***"Compared to DT or GDT, Skill DT doesn't have actions as inputs. Is there any justification/explanation?"***
>
> We have an explanation for this in the paper – “we omit actions in predictions. This is because we are interested in SMM through skills, where we want to extract as much information from states.“ Similar to works such as EDL, we want to rely solely on state information to perform SMM.
>
> ***"How does skill DT prepare the future skill distribution in the evaluation time?"***
>
> We’ve added some detailed python pseudocode in the appendix explaining how to do this.  To evaluate a skill, we first populate a buffer of skills. When we rollout in the actual environment, the causal transformer utilizes this buffer to make predictions. At every step, Skill DT updates the skill buffer with the encoded actual state that it observes.
>
> ***"Continuous skill-based methods often assume gaussian distribution as a skill distribution, which is easy to sample latent skills. It should be clarified how "easily sampling diverse behaviors" in Skill DT. Since Skill DT seems to sample the discrete skill variable at every timestep, it seems difficult to sample desired / consistent behaviors."***
>
> It's possibly to sample skills with a standard gaussian, but it becomes challenging to evaluate supervised return in an unsupervised manner. It becomes challenging to figure out which continuous skill results in the highest return in an environment (taking multiple random samples may not cover the skill space enough, and could result in more overlapping behaviors). Works like OPAL utilize bypass this by including a hierarchical policy that uses learned skill embeddings and is trained using *supervised* rl. In our case, evaluating learned skill space is trivial because we have a discrete set and can just perform an environment rollout for each skill individually.
>
> ***"There is no skill-based baseline. GDT-variants (CDT/BDT), k-means clustering-conditioned DT, or some offline skill-based method (OPAL, LiSP [1]) can be relevant baselines."***
>
> We’ve added OPAL and K-Means augmented Decision Transformer as a skill based comparison

---

> > ### Comment · Reviewer_L7Nj · 2022-12-09
> > **Response**
> >
> > I thank the author for the update of your manuscripts and detailed response. The paper seems better than before, but I still find several points that should be fixed remains (for example, Figure 1 should use context length instead of $T$. Actually, $T$ seems to be an undefined variable. The GDT-like formalization of Skill DT in Section 4.1 might be wrong: $I = (histogram, z_t)$ --> $I = histogram$?, or (histogram, z_t) doesn't make sense. Why are histogram and z_t separately treated, while histogram includes the information of z_t?. Moreover, I'm concerned that the originality of this paper is very limited to the replacement of the state-discretization / anti-causal transformer encoder with VQ-VAE encoder. For the reasons above, I'd like to maintain my rating as it is.

---

### Official Review · Reviewer_Tnhd · 2022-10-24

**Confidence:** 4
**Correctness:** 2
**Technical Novelty And Significance:** 3
**Empirical Novelty And Significance:** 2
**Recommendation:** 3

**Clarity, Quality, Novelty And Reproducibility:**

Clarity: Overall the presentation is quite clear, however there is no description of the VQ-VAE training procedure.
A further (minor) question: the formulas for computing the skill distribution do not include normalization, but the diagrams in Figure 4 suggest that the resulting histogram is normalized?

Quality: Given the missing experiments mentioned above, I don't think the paper fully supports the proposed architecture.

Novelty: The building blocks for the architecture stem from prior work: Decision Transformers and state partitioning for skill discovery. The combination is novel but not sufficiently evaluated.

Reproducibility: Details regarding the VQ-VAE training procedure are missing and source code is not provided; as such, I don't think that work could be reproduced as is.

**Strength And Weaknesses:**

Strengths:
- Decision Transformers are a recently introduced paradigm, and further developments on RL and sequence modeling are highly relevant to the community, as are skill discovery approaches.
- The paper is generally easy to follow.
- The proposed model can recover high-performing trajectories on benchmark offline RL datasets, although several questions remain open (see below)

Weaknesses:
- The method is not fully described. How is the VQVAE learned that s used to quantise states in Algorithm 1? Why is another encoder learned? Is the algorithm working with a random code-book? Why not use k-means instead then?
- I think the evaluation here is somewhat unfair. On the one hand, your method can act in a zero-shot manner, but on the other hand you have to try every single skill and then pick the top-performing one. This should be made clear in the table, e.g., by relabeling the "Skill DT" column as "Best Skill DT". What's the average and worst performance of the skills?
- The experiments leave several questions unanswered which would be very helpful to better understand the method:
  - What do the resulting skills look like? Figure 3(a) only shows rollouts with a fixed skill, right? Given that this is a maze and that X/Y coordinates are part of the state, is there no correlation between skill partition and X/Y position at all? If it's not X/Y, what do the skills represent then? Or, put differently, what do a state at the start and end of a trajectory have in common so that they end up in the same cluster? It's hard for me to see how single-state clustering can be used to describe full trajectories consistently.
  - Figure 5 and 6 are hard to interpret. The conclusion seems to be that a clustering is learned, but it's hard to judge the quality here. The VQVAE (for which it's unclear how it's learned) is described to perform better than k-Means. Why not try k-Means in the full pipeline instead to get a more grounded comparison?
  - What is the quantitative benefit of providing codebook vectors rather than one-hot vectors for the active skill in a specific state?
  - What is the relation between the number of skills and final performance or behaviors learned?
- In Figure 4, results indicate that the policy is able to replay a reference trajectory. Would that not also work with the vanilla Decision Transformer, or another GDT variant that takes future statistics (e.g., simply the next 5 states) into account? You say "[SDT] is able to follow the general path in a zero shot manner", but is there any trajectory in the dataset that does *not* go from the start to the target?
- No videos of acting policies provided.

**Summary Of The Paper:**

The paper introduces the Skill Discovery Decision Transformer, an extension of Decision Transformer [1] in which that does not use the return-to-go for policy conditioning. Instead, individual states are clustered and conditioning is performed with a "skill-distribution-to-go". The general motivation is a relation to (unsupervised) skill discovery algorithms which often do state-space clustering in order to learn goal-conditioned low-level policies (e.g., implicitly as DIAYN [2] or explicitly as EDL [3], OPAL [4]). However, in this paper, the policy is modeled with a transformer rather than an MLP.

[1] http://arxiv.org/abs/2106.01345
[2] http://arxiv.org/abs/1802.06070
[3] http://arxiv.org/abs/2002.03647
[4] https://openreview.net/pdf?id=V69LGwJ0lIN

**Summary Of The Review:**

While the overall idea of the work at hand is interesting, I don't see it being fit for publication in the current form. The overall algorithm is not fully described, and several important analyses and ablations are missing. Pages 7 and 8 are populated with Figures that could well be placed in the Appendix, making room for more relevant additions.

---

> ### Author Response · Authors · 2022-11-19
> **Response to Reviewer Tnhd**
>
> Thank you reviewer Tnhd for the great feedback! We'd like to address some of your comments:
>
> ***"The method is not fully described. How is the VQVAE learned that s used to quantise states in Algorithm 1? Why is another encoder learned? Is the algorithm working with a random code-book? Why not use k-means instead then?"***
>
> We added a new figure ( Figure 2 ), which shows a single training iteration. The VQVAE is trained concurrently with the causal transformer, trained via action reconstruction loss. We also took your suggestion and have added a new baseline – K Means Decision Transformer. The K Means Decision Transformer clusters states, and uses cluster centers as skill embeddings.
>
> ***"I think the evaluation here is somewhat unfair. On the one hand, your method can act in a zero-shot manner, but on the other hand you have to try every single skill and then pick the top-performing one."***
>
> Evaluating Supervised return and few shot SMM both have different procedures with Skill DT. Few shot SMM only requires one rollout, given by the skill encodings and skill histograms calculated by the trajectory. However, because Skill DT is purely unsupervised, we have to iterate through the skills to evaluate. In future work we want to introduce some hierarchical RL / supervised learning methods to guide Skill DT.
>
> ***"What do the resulting skills look like? Figure 3(a) only shows rollouts with a fixed skill, right? Given that this is a maze and that X/Y coordinates are part of the state, is there no correlation between skill partition and X/Y position at all? If it's not X/Y, what do the skills represent then? Or, put differently, what do a state at the start and end of a trajectory have in common so that they end up in the same cluster? It's hard for me to see how single-state clustering can be used to describe full trajectories consistently."***
>
> Because Skill DT encodes the full states, encoded skills in an environment like ant-umaze don’t necessarily represent just x / y coordinates. We interpret the skills as encoding specific trajectory paths. Even though the VQVAE encodes single states at a time, because it is updated through backpropagation through the causal transformer, which predicts actions using multiple sub trajectories of states. We also show TSNE projected ant skill embeddings  (Figure 4), which shows that Skill DT is able cluster states into well defined regions. In addition, we’ve also included images of different skill trajectories in the appendix.
>
> ***"The VQVAE (for which it's unclear how it's learned) is described to perform better than k-Means. Why not try k-Means in the full pipeline instead to get a more grounded comparison? What is the quantitative benefit of providing codebook vectors rather than one-hot vectors for the active skill in a specific state?"***
>
> We’ve included a K-means Decision Transformer baseline, which performs better than the original Decision Transformer, but performs worse than Skill DT.
>
> ***"You say "[SDT] is able to follow the general path in a zero shot manner", but is there any trajectory in the dataset that does not go from the start to the target?"***
>
> The ant-umaze-diverse dataset has a very small number of trajectories that are actually successful, as we show in our dataset statistics table (the average normalized reward is 1.2, max is 100, min is 0). In addition, the trajectory we chose is unique in that it contains a specific loop in the top left corner of the maze, which other trajectories don’t have.
>
> ***"What is the relation between the number of skills and final performance or behaviors learned?"***
>
> We’ve added an ablation study comparing the performance of a range of number of skills for each dataset / environment. For environments with unimodal behaviors (-medium), the number of skills has less of an effect on performance. However, for (-replay) environments, its apparent that increasing number of skills helps improve performance.

---

### Official Review · Reviewer_2Y2V · 2022-10-25

**Confidence:** 4
**Correctness:** 3
**Technical Novelty And Significance:** 2
**Empirical Novelty And Significance:** 2
**Recommendation:** 5

**Clarity, Quality, Novelty And Reproducibility:**

The paper is in good clarity, of decent quality, and relatively borderline novelty. The reproducibility of the paper should be good.

**Strength And Weaknesses:**

Strength:
+ The idea is clearly stated, and the implementation is relatively simple based on the well-established components such as VQ-VAE and Decision Transformers;
+ The proposed method achieves improved performance across a large number of domains in D4RL benchmarks;

Weaknesses
- The proposed idea is rather incremental based on the decision transformers and generalized decision transformers, where I don't see many specific technical contributions other than combining VQ-VAE to have a discrete skill set;
- The number of skills set in the proposed method seems rather arbitrary from Table 1. How should the users specify how many skills the proposed method should employ for better performance?
- The illustrations of Figure 5 and Figure 6 do not seem to add much information for understanding the method;
- I would expect an experiment, where sampling and conditioning on different skills the trajectory could generate different styles/behaviors;
- The proposed method should be more effective in tasks requiring hierarchical RL in complex control domains, where low-level skills are discovered in an unsupervised fashion.


**Summary Of The Paper:**

The paper extends Decision Transformers and Generalized Decision Transformers for skill discovery. They learn primitive skills based on unsupervised learning (i.e., without rewards information). Technically, they first use a VQ-VAE to encode discrete skills, then they utilize a causal transformer just like the original decision transformer. For training, they condition the encoded skill,  the distribution of skills, etc, to predict the actions. In experiments, they use D4RL benchmark, where they show the skill decision transformer could leverage the unsupervised-learned skills to improve the performance.

**Summary Of The Review:**

The paper combines VQ-VAE and decision transformers to tackle the important problem of skill discovery. However, the experiments could be improved to demonstrate the strength of such a neat combination.

---

> ### Author Response · Authors · 2022-11-19
> **Response to Reviewer 2Y2V**
>
> Thank you reviewer 2Y2V for the great feedback! We'd like to address some of your comments:
>
> ***"The proposed idea is rather incremental based on the decision transformers and generalized decision transformers, where I don't see many specific technical contributions other than combining VQ-VAE to have a discrete skill set"***
>
> Our approach is an extension of the decision transformer and generalized decision transformer variant, which uses a VQ-VAE as a discrete skill set. However, we utilize these skill in a novel manner by generating future skill distributions (skill histograms) which capture trajectory information. In addition, even though our approach is completely unsupervised, we are able to outperform other state of the art offline RL methods on a number of tasks.
>
> ***"The number of skills set in the proposed method seems rather arbitrary from Table 1. How should the users specify how many skills the proposed method should employ for better performance?"***
>
> We’ve added an ablation study comparing the performance with a varying number of skills. It seems that for datasets with multimodal behaviors (-replay environments, antmaze-diverse environments) show increased performance with a larger number of skills. This is because when we evaluate Skill DT, we have to try every single skill to determine the best reward achieved. With datasets that contain diverse behaviors, Skill DT may need more skills to accurately cover the behavior space of the dataset. In general, more skills can improve performance, but there’s a tradeoff in compute time because it requires more rollouts with each skill.
>
> ***“I would expect an experiment, where sampling and conditioning on different skills the trajectory could generate different styles/behaviorsI would expect an experiment, where sampling and conditioning on different skills the trajectory could generate different styles/behaviors”***
>
> We’ve included some images of skill trajectories in the appendix showing different behaviors. We've also plotted skill embedding clusters to show that Skill DT is able to cluster states into distinct skill regions.
>
> ***“The proposed method should be more effective in tasks requiring hierarchical RL in complex control domains, where low-level skills are discovered in an unsupervised fashion.”***
>
> We agree with this, and believe that this is a good future direction for our work. In this paper however, was choose to focus exclusively on unsupervised offline skill discovery.

---

> > ### Comment · Reviewer_2Y2V · 2022-12-09
> > **Ack of the response**
> >
> > Thank the authors for their efforts to make the paper better. However, I think the paper is still below the publication bar of ICLR conference. Thus I'll keep my evaluation score.

---

### Official Review · Reviewer_tKoW · 2022-11-04

**Confidence:** 3
**Correctness:** 3
**Technical Novelty And Significance:** 2
**Empirical Novelty And Significance:** 2
**Recommendation:** 5

**Clarity, Quality, Novelty And Reproducibility:**

- Clarity: Good.
- Quality: Fair.
- Novelty: Poor. This work is just built on previous works, such as Decision Transformer and  Skill Discovery
- Reproducibility: Poor. The authors provide no code, no appendix, and no hyper-parameter description.

**Strength And Weaknesses:**

Strength:
1. This paper is well-writing.
2. The discovery of diverse behaviors is an interesting problem.

Weakness:
1. The experiment results are not good enough. The proposed method is much worse than baselines in some tasks (such as antmaze-medium and antmaze-mediumdiverse).
2. Lack of diverse behavior analysis. (1) The author should give some visualization of diverse behaviors their method finds on the D4RL dataset. (2) The author should give a diversity metric to measure the diversity of behaviors on different tasks.


**Summary Of The Paper:**

This paper proposed a transformer-based architecture to discover diverse behaviors from the offline dataset.

**Summary Of The Review:**

This paper proposed a transformer-based architecture to discover diverse behaviors from the offline dataset. The paper is somewhat incremental work and with unsatisfied experimental evaluation.

---

> ### Author Response · Authors · 2022-11-19
> **Response to Reviewer tKoW**
>
> Thank you reviewer tKoW for your great feedback! We'd like to address a couple of your comments:
>
> ***"The experiment results are not good enough. The proposed method is much worse than baselines in some tasks (such as antmaze-medium and antmaze-mediumdiverse)"***
> The methods that perform well on these tasks (CQL, IQL, OPAL) utilize either dynamic programming or hierarchical reinforcement learning utilizing supervised returns. However, we don’t use any dynamic programming or supervision, but we still do better than the non-dynamic programming baselines (DT, K means DT).
>
> ***"Lack of diverse behavior analysis. (1) The author should give some visualization of diverse behaviors their method finds on the D4RL dataset."***
>
> We’ve added trajectory sequences for different skills and different tasks in the appendix.

---

### Author Response · Authors · 2022-11-19
**Summary of Rebuttal Changes**

 We’d like to thank all the reviewers for the great feedback! There were a lot of very useful suggestions, hopefully we have addressed all of them with the new revision. List of updates:

- Added additional skill baselines – OPAL, KMeans Decision Transformer. KMeans Decision Transformer utilizes K means clustered states instead of the trained VQVAE Skill DT uses.
- Added a new diversity metric utilizing Wasserstein loss between skill encoded trajectories
- Added ablation study comparing performance across different number of skills
- We’ve added an additional diagram visualizing a training iteration to make it a bit clearer on how the different components are trained.
- We’ve added an addition table showing offline dataset information
- We’ve added sudocode describing the evaluation process
- We’ve added trajectory images for multiple environments.
- We’ve also published our code here: https://anonymous.4open.science/r/skill-dt-8048/

---

### Decision · Program_Chairs · 2023-01-20

**Decision:**

Reject

**Justification For Why Not Higher Score:**

Very low novelty given prior work.

**Justification For Why Not Lower Score:**

N/A

**Metareview: Summary, Strengths And Weaknesses:**

The paper proposes an extension of Decision Transformers to condition on skills. The original model [Chen et al.] conditions on returns, whereas another extension Generalized DT conditions on arbitrary statistics. The current works extends this to conditioning on skills. While the reviewers appreciated the clarity of the work, there were uniform concerns among the reviewers on the limited novelty of the work (changing the conditioning to skills and use of VQ-VAE encoder) and the sparse details of the empirics. Following up one of the reviewer suggestions, I'd encourage the authors to investigate hierarchical RL environments for further investigation and analysis of the proposed approach.